# Normalization of Language Embeddings for Cross-Lingual Alignment

## Abstract

Learning a good transfer function to map the word vectors from two languages into a shared cross-lingual word vector space plays a crucial role in cross-lingual NLP. It is useful in translation tasks and important in allowing complex models built on a high-resource language like English to be directly applied on an aligned low resource language. While Procrustes and other techniques can align language models with some success, it has recently been identified that structural differences (for instance, due to differing word frequency) create different profiles for various monolingual embedding. When these profiles differ across languages, it correlates with how well languages can align and their performance on cross-lingual downstream tasks. In this work, we develop a very general language embedding normalization procedure, building and subsuming various previous approaches, which removes these structural profiles across languages without destroying their intrinsic meaning. We demonstrate that meaning is retained and alignment is improved on similarity, translation, and cross-language classification tasks. Our proposed normalization clearly outperforms all prior approaches like centering and vector normalization on each task and with each alignment approach.

## 1  Introduction

The best multilingual NLP approaches typically do not jointly learn a single embedding, since words of the same language tend to cluster, and thus are not useful for translation and cross-lingual learning tasks. Rather, after learning individual embeddings, the standard approach is to map word vectors from multiple languages into a shared cross-lingual word vector space [15]. This shared space creates a cross-lingual word embedding (CLWE) [22, 43]. These serve as a valuable tool for transferring data across different languages, understanding cross-linguistic differences, and cross-lingual transfer for downstream tasks, such as direct translation [16, 20, 24], cross-lingual information retrieval [42], cross-lingual document classification [23], and cross-lingual dependency parsing [17, 39].

A common element of almost all CLWE methods is the use of a rigid, orthogonal transformation mapping one embedding onto another so they inhabit a shared linguistic space. An orthogonal transformation is a special class of transformations that can be interpreted as the space of (in our case, high-dimensional) rotations around the origin, and also allowing a mirror flip. This family of transformations preserves (a) linear and (b) angular properties. By linear properties, we mean that the straight-line Euclidean distance between elements is preserved, as are more powerful properties like analogies (e.g., Paris - France + Italy $\approx$ Rome). Angular properties refer to measuring angles between pairs of points (from the origin), and as a result, cosine distance is preserved. Given a correspondence between pairs of objects across two embeddings, the classic Procrustes method, provides a closed-form solution which minimizes the sum of Euclidean distances. Moreover, if the vectors are all first made as unit vectors, then this also maximizes the sum of cosine similarities [10].

Submitted to 35th Conference on Neural Information Processing Systems (NeurIPS 2021). Do not distribute.

Under this framework, there has been a flurry of work significantly improving CLWE model performance along two directions. Semi-supervised and unsupervised models make these approaches require less input, and more amenable to lower-resource languages. For example Bootstrap Procrustes (PROC-B) [15, 41] is semi-supervised in that it starts with a small pairwise correspondence (of 500-1000 words), aligns those to infer a larger correspondence, and repeats applying Procrustes alignment. Methods like MUSE [8] are unsupervised, and use a GAN to estimate a correspondence before applying a Procrustes procedure.

The second direction is preprocessing the embeddings before applying the Procrustes alignment. These involve methods like removing the mean, removing principal components, and normalization which we will discuss in depth later. In principle, these methods aim to remove the geometry of data intrinsic to particular languages (but not shared across languages) while preserving similarity properties as assured by orthogonal alignment. The space of transformations allowed under orthogonal alignments is quite large, and we make the point that unless this data geometry is "normalized" it inhibits the alignment from optimizing over the entirety of this large space.

Finally, we note that methods like Canonical Correlation Analysis (CCA) [13], Discriminative Latent Variable (DLV) [36], and Ranking-based optimization (RCSLS) [21] have also been applied towards finding an orthogonal alignment (or pair of alignments) which minimizes a different optimization function – since the objective function may not align with sums of squared Euclidean of cosine distance [8, 38]. Unlike the others, the RCSLS method notably does not require a rigid transformation.

The focus of this paper is on embedding *preprocessing*, and is agnostic to the method of alignment used afterward, whether it is Procrustes-based, or optimizing something else.

**Our contribution.** This work proposes a new and general approach to preprocessing word embeddings, subsuming many previous approaches. The key is *Spectral Normalization* which regularizes the spectral properties of monolingual embeddings by setting all of the top singular vectors to have the same singular value. However, it leaves alone the smaller singular value; these capture important information and cannot be zeroed out, but making them the same value as the top singular vectors introduces too much noise. Spectral normalization already performs as well as the best previous approaches on alignment and translation tasks, and since it applies a fairly uniform stretching to the embeddings it does not distort monolingual similarity performance. Moreover, we show layering Spectral Normalization within an iterative sequence with also centering and vector length normalization improves results further. We first demonstrate this improvement on the standard translation task (BLI). We also show that this normalization preserves the core ontological structure of embeddings across languages, and that applying our normalization before aligning a low resource language to English improves performance on topic classification and on a natural language inference task.

## 2 Existing Methods for Orthogonal Vector Spaces Alignment

Given a language $L$, our starting point is an embedded representation of a set of $n$ words. Indexing words from $i = 1 \ldots n$, each is associated with a vector $x_L^i \in \mathbb{R}^d$. And let $X_L = \{x_L^1, \ldots, x_L^n\}$ be the set of $n$ words as their vector representation. These vector representations (derived by methods like word2vec [28], GloVe [33], or FastText [4]) are chosen so words with similar pairwise cosine similarity are found in the similar local context in large text corpora on which they are trained. Higher-level linear structure is shown to emerge, such as concept subspaces and analogies [29].

The focus of this paper is on aligning embeddings of two languages $L_1$ and $L_2$. Each embedding $X_{L_1}$ and $X_{L_2}$, only is designed to ensure pairwise relationships between its word vectors, but the actual coordinates of those vectors do not have any explicit meaning. Yet, previous work has clearly demonstrated that there exists significant overall structural similarity, and alignment seeks to make correspondences between those structures for translation and joint understanding.

Most methods start with a known correspondence (or build one) between a set of $K$ words in two languages, wlog let these be the same first $K$ indexed words in those languages, denoted $X_{L_1}^K = \{x_{L_1}^1 \ldots x_{L_1}^K\}$ and $X_{L_2}^K = \{x_{L_2}^1 \ldots x_{L_2}^K\}$. Then the *Procrustes Problem* solve for an orthogonal matrix $W^* = \arg\min_W \|X_{L_1}^K W - X_{L_2}^K\|_2^2$. There exists a simple solution [2, 38, 45, 10] as $W^* = UV^\top$ where $U\Sigma V^\top = \mathsf{svd}(X_{L_1}^K (X_{L_2}^K)^\top)$. Dev *et.al.* [10] also point out that if all vectors are normalized first, then this procedure also maximizes the sum of cosine similarities.

## 2.1 Pre-processing Embeddings before Orthogonal Alignment

It turns out directly aligning embeddings from two languages (even using the "optimal" Procrustes solution) does not provide the best possible joint embedding for translation tasks. While word meaning appears to hold a similar structure, languages have other properties such as differing word frequency, and this for instance leads to more frequent words having longer vectors in embeddings. This extra language-specific structure tends to interfere with alignment. As a result, a number of techniques have been developed to "normalize" the embeddings before Procrustes (or other) alignment. This in some sense allows the word meaning to dominate the optimization tasks without other confounding factors. We review the most common normalization approaches.

**Mean Centering (C)** subtracts the mean of all vectors in an embedding from each vector in that embedding. The result is that the mean of all vectors is 0. This is a rigid transformation, and so does not change the Euclidean distance between any pair of points in an embedding, and also preserves any linear property like analogies (e.g., Paris - France + Italy ≈ Rome). Dev *et. al.* [10] points out that this is the first step (followed by the Procrustes orthogonal transformation) to minimize the sum of squared Euclidean distances among paired words, under any rigid transformation. However, this *does change* the cosine distance between pairs of points.

**Length Normalization (L)** makes each vector have a 2-norm equal to 1, but retains its direction from the origin [2, 45]. This preprocessing step does not change the cosine distance between any pair of points in an embedding. But, it *does change* the Euclidean distance between pairs of points.

Despite these contrasting goals, these two normalizations each turn out to be individually effective in regularizing the geometry of the embeddings, and allow for better CLWE. Wang *et. al.* [47], realized doing both was even more effective, and showed that iterating these two steps achieves the state-of-the-art way to preprocess, we denote as **I-C+L**. Iterative Normalization transforms monolingual word embeddings to have unit-length and zero-mean simultaneously (in practice they terminate this iterative process after a few steps before it achieves these two goals exactly).

**PCA Removal (PR)** computes the principal component analysis (PCA) of an embedding, and then projects away from the direction of the top principal component, removing it [31]. Mu *et. al.* [31] observed that the top singular values typically do not encode essential semantic relationships between words but rather align strongly with word frequency. Also, they notice that the top principal values are much larger than the other values. After eliminating $d/100$ principal components, where d is the dimension of a word representation, they achieve better performance on both intrinsic and extrinsic tasks. Sachidananda *et. al.* [37] applied this simple pre-processing approach to their Filtered Inner Product Projection (FIPP) alignment method, and they significantly improve on the BLI task.

## 2.2 Spectral Statistics of Embeddings

Dubossarsky *et. al.* [11] recently documented how cross-lingual alignment is strongly affected by the spectral statistics of monolingual embeddings. We stack the embedded vectors $x_L^i \in \mathbb{R}^d$ as rows in a $n \times d$ matrix $A \in \mathbb{R}^{n \times d}$. The SVD decomposes $A$ into $U \Sigma V^\top$ where $U$ and $V$ contain the left and right singular vectors, and the singular values $\sigma_1 \geq \sigma_2 \geq \ldots \geq \sigma_d \geq 0$ are on the diagonal of $\Sigma$. The *effective rank* of $A$ is a smoother analog to rank (when there is noise in low rank components), defined $\mathrm{er}(A) = e^{H(\Sigma)}$ where $H(\Sigma) = -\sum_{i=1}^d \bar{\sigma}_i \log \bar{\sigma}_i$ with $\bar{\sigma}_i = \sigma_i / \sum_{i=1}^d \sigma_i$. The *effective condition number* $\kappa_{\mathsf{eff}}(A) = \sigma_1 / \sigma_{\mathrm{er}(A)}$, which replaces the numerator (of condition number, $\sigma_d$) with the more robust singular value at the effective rank. This is desired to be small in stable data sets. The *joint effective condition number* measures the harmonic mean of the effective condition number across two matrices $A, A'$ as $\mathrm{ECOND\text{-}HM}(A, A') = \frac{2\kappa_{\mathsf{eff}}(A)\kappa_{\mathsf{eff}}(A')}{\kappa_{\mathsf{eff}}(A)+\kappa_{\mathsf{eff}}(A')}$. The *singular value gap* measures how similar the singular value sequences are between two matrices as $\mathrm{SVG}(A, A') = \sum_{i=1}^d (\log \sigma_i - \log \sigma_i')^2$. These should both be smaller, for more comparable data sets.

Dubossarsky *et. al.* [11] applied these to monolingual embeddings and demonstrated that the performance of several CLWE methods were closely tied to these spectral properties. Basically embeddings align better if they are better jointly conditioned, especially measured via joint effective condition number and the singular value gap. Motivated by this idea, we propose methods that spectrally normalize embeddings improving these statistics while retaining intra-embedding meaning.

## 3 New Normalization Methods

We introduce more direct and more general techniques to normalize monolingual word embeddings to more effectively prepare them for alignment. The goal is to remove language-specific geometry while maintaining the intrinsic similarity and structure captured within them.

### 3.1 Geometric Median Normalization

Iterative Normalization enforces individual word embeddings to have a unit length and each monolingual embedding to have a zero mean through an iterative technique. Wang *et. al.* [47] showed that iterating solutions for these distinct goals will eventually converge to a solution which satisfies both.

In this paper, we observe that both goals can be done in one shot without iterating – by solving the Fermat-Weber problem [27, 25]. This dates to the 17th century, and corresponds with identifying the geometric median of a point set. Formally, the goal is a point $x^* \in \mathbb{R}^d$ that minimizes the sum of distances from $n$ anchor points $\{a_1, \ldots, a_n\} \subset \mathbb{R}^d$ which are not collinear: $x^* = \min_x \sum_{i=1}^n \|x - a_i\|$. Several methods [6, 12, 32] have been proposed; the most popular is the Weiszfeld's algorithm (Weiszfeld, see Appendix A). It is folklore that the solution $x^*$ satisfies that $0 = \sum_{i=1}^n \frac{a_i - x^*}{\|a_i - x^*\|}$; we do not know of a written proof, so prove this in Appendix A.1 for completeness.

Using this characteristic of the geometric median, we can simultaneously enforce monolingual word embeddings to have unit-length and zero-mean in just one step. This can be done using the Geometric Median normalization (GeoMediaN) algorithm (as Algorithm 1). Given a monolingual word embedding $A$, we compute the geometric median $x^*$, and "center" the data on this point, and unit length normalizes the centered embedding.

---

**Algorithm 1** Geometric Median Normalization: GeoMediaN($A$)

1: $x^* \leftarrow$ Weiszfeld($A$)
2: **for all** $a_i \in A$ **do** $a_i \leftarrow \frac{a_i - x^*}{\|a_i - x^*\|}$
3: **return** $A$

---

After these steps, all vectors are unit length, and because of the folklore property (Theorem A.1), the mean of those points is also $0$. As a result, we can state the following property.

**Theorem 3.1.** *The output of* GeoMediaN($A$) *is centered and length normalized.*

Despite the Geometric Median Normalization algorithm's ability to enforce unit-length and zero-mean in just one step, we will observe that it does not perform especially well on the BLI task. Both GeoMediaN and I-C+L achieve one of many solutions which achieve these joint goals. We next investigate another one that works better: it preserves meaning and structure, and removes language-specific geometry allowing improved alignment.

### 3.2 Spectral Normalization

The predominant effect of unit-length and zero-mean normalization on monolingual word embeddings is that it makes embedding vectors from a language lie on a hypersphere with the center of the hypersphere centered at the origin. However, this does not take into account how the word embeddings vectors are spread out or clustered on the hypersphere. Approaches like PCA removal and mean centering have the effect of reducing the top principal component or top singular vector. As a result, if the spectral properties are extreme, it can help regularize them. However, this approach can be a bit blunt. PCA removal makes the top singular value exactly $0$, so the condition number becomes infinite. Other quantities like the effective condition number, $\kappa_{\text{eff}}(A)$, do however, tend to decrease.

To this effect, we propose a new algorithm Spectral Normalization that more gently regularizes the spectral properties of word embeddings; see Algorithm 2. We will then combine it with other approaches to again ensure the embedding vectors lie on the unit sphere.

---

**Algorithm 2** Spectral Normalization ($\mathsf{SpecNorm}(A, \beta)$)

---

1: Compute $\mathsf{svd}(A) = U\Sigma V^\top$; Let $D \in \mathbb{R}^d$ be a diagonal matrix.
2: Compute $\eta = \sqrt{\|A\|_F^2/d}$, where $d$ is the dimension of the word embedding
3: **for** $i = 1, \ldots, d$ **do**
4:     **if** $(\Sigma_{ii} > \beta\eta)$ **then** $D_{ii} \leftarrow \Sigma_{ii}/(\beta\eta)$
5:     **else** $D_{ii} = 1$.
6: **return** $AVD^{-1}$

---

Given a monolingual word embedding $A$ it updates part of the spectral properties of $A$ as a whole, using on a parameter $\beta \geq 0$. Based on an average of singular values $\eta = \sqrt{\|A\|_F^2/d}$, if a value is above $\beta$ times that average, it adjusts it to $\beta\eta$. Hence, all of the top directions are given the same singular value. Otherwise, if it is below $\beta\eta$, it is considered a minor effect (some are quite small, and fairly noisy), and it is left alone. If these small ones are completely zeroed out, the critical information within is destroyed. However, if these small ones are also given the same value (i.e., $\beta\eta$) then components which do not contribute to the most prevalent aspects of a vectors similarity is given more importance, and we observed (see Section 4.1) that the usefulness of the embedding decreased.

**Iterative Spectral Normalization.** Spectral normalization makes the most sense (see Appendix G) in a setting where the vectors are already centered, and also unit length. While SpecNorm does not change the center of the data, it does not maintain the length of individual vectors. As such, we advocate combining these methods into a single iterative algorithm: I-C+SN+L as in Algorithm 3.

---

**Algorithm 3** Iterative Spectral Normalization with C+L normalization (I-C+SN+L$(A, \text{\#Iter})$)

---

1: **for** #Iter steps **do**
2:     $A \leftarrow$ Center $A$
3:     $A \leftarrow \mathsf{SpecNorm}(A)$
4:     $A \leftarrow$ Unit length normalization of $A$
5: **return** $A$

---

We observe in Figure 1 that this process significantly improves the spectral properties, compare to any other approach. Without preprocessing (None), the languages (EN: English, DE: German, HI: Hindi, JA: Japanese shown) have large effective condition numbers – indicating that there is a large disparity between meaningful singular values. Note the y-axis is in log scale. Hence, aligning these languages without normalization would likely restrict alignment among top singular vectors, not allowing enough degree of freedom to align corresponding words.

In contrast, after preprocessing when these values are more uniform, rotations among the dimensions containing the top principal components will not have an influence on the data distribution, and can fully optimize the alignment between words. Moreover, Figure 1 shows that I-C+SN+L most decreases the effective condition number, joint effective condition number, and singular value gap. Further, these values are fairly uniform across languages, despite great variation beforehand (as shown with None). In fact, I-C+SN+L is much more effective than other methods.

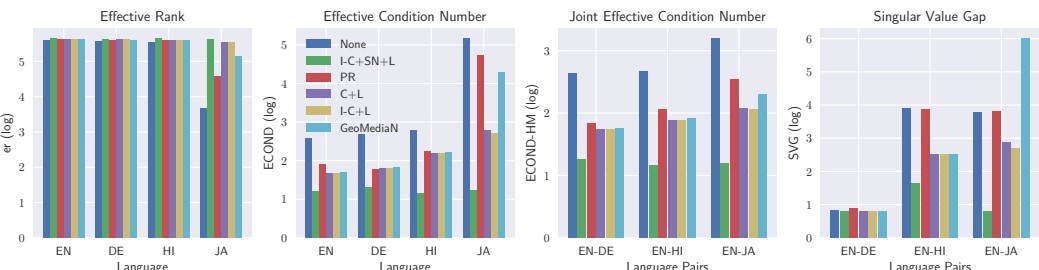

Figure 1: Spectral Measures of four (4) monolingual word embeddings, before (None) and after applying various normalization methods.

## 4 Experimental Analysis

We provide an evaluation of our proposed preprocessing methods using 8 language embeddings pretrained on Wikipedia [4] of each language: Croatian (HR), English (EN), Finnish (FI), French (FR), German (DE), Italian (IT), Russian (RU), and Turkish (TR). We use the 300-dimensional fastText [4][1] embeddings, and all vocabularies are trimmed to the 200K most frequent words.

**Alignment evaluation tasks: BLI** We evaluate and compare our proposed preprocessing methods mostly on the Bilingual Lexicon Induction (BLI) task, a word translation task. We discuss two more global applications Cross-lingual document classification (CLDC), and Cross-lingual transfer for natural language inference (XNLI) later. BLI is more direct, and has become the de facto evaluation task for CLWE models. For words in the source language, this task retrieves the nearest neighbors in the target language after alignment to check if it contains the translation. It reports the mean average precision (MAP) [15], which is equivalent to the mean reciprocal rank (MRR), of the translation. Unless stated otherwise, reported values on baseline methods are taken from [15], and use the Google Translate (GTrans) dictionary from [15][2]. We trained (aligned) using 1k, 3k and 5k source words and evaluated (tested) on separate 2k source test queries, unless noted otherwise.

**Alignment Algorithms.** We evaluated and compared the result of several supervised rigid-transformation CLWE models on the evaluation benchmarks using our proposed methods. All have publically available code, links are found in the reference citation. These include Canonical Correlation Analysis (CCA) [13], Procrustes (PROC) [2, 38, 45, 10], Bootstrapping Procrustes (PROC-B) [15], and Discriminative Latent-Variable (DLV) [36], as discussed in Section 1. We also consider Ranking-Based Optimization (RCSLS) [21] which is not a rigid alignment. In a few places, we also compare with VECMAP [3] as an example of an unsupervised alignment process. This should only use the geometry of the global embedding structure, e.g., derived from the natural ontology, and our normalization method still helps when using this approach.

### 4.1 Hyperparameter Tuning

Our main proposed algorithm I-C+SN+L has a few simple parameters. To avoid overfitting, we choose these through cross-validation on English (EN) and a held-out set of 5 languages Hindi (HI), Russian (RU), Chinese (ZH), Japanese (JA), Turkish (TR). Ten (10) Language pairs of the form EN-X and X-EN were considered. The hyperparameters $\beta \in \{1, 2, 3, 4, 5\}$ and #Iter (number of iterations) $\in \{1, 2, 3, 4, 5\}$ were fine-tuned for I-C+SN+L.

We used the publicly available MUSE[3] translation dictionary [8] for hyperparameter tuning. The Procrustes alignment algorithm was trained on 5k source words and evaluated on 1.5k source test queries. We reported the mean average precision (MAP) in Table 1 for $\beta \in \{1, 2, 3, 4, 5\}$ and with #Iter $\in \{1, 2, 3, 4, 5\}$. We observe the value of $\beta = 2$ was consistently the best threshold (although any $\beta \geq 2$ performed similarly). These singular values were normalized, and those below $\beta$ times the average we judged as noise, and left as is. However, the result did not change much with respect to the number of iterations.

Table 1: Cross-Validation for Hyperparameter Tuning: MAP after Procrustes for 10 language pairs.

| $\beta$ | #Iter=1 | #Iter=2 | #Iter=3 | #Iter=4 | #Iter=5 |
|---|---|---|---|---|---|
| 1 | 0.363 | 0.340 | 0.328 | 0.322 | 0.317 |
| 2 | 0.385 | **0.386** | **0.386** | **0.386** | **0.386** |
| 3 | 0.381 | 0.384 | 0.384 | 0.384 | 0.384 |
| 4 | 0.381 | 0.382 | 0.382 | 0.382 | 0.382 |
| 5 | 0.380 | 0.381 | 0.381 | 0.381 | 0.381 |

The tie between the #Iter hyperparameter was broken using their performance on thirteen English word similarity benchmarks; see Appendix C for more details. In Table 2 below, we report the

---

[1]`https://github.com/facebookresearch/fastText`

[2]`https://github.com/codogogo/xling-eval?utm_source=catalyzex.com`

[3]`https://github.com/facebookresearch/MUSE`

average Spearman rank coefficient score on the word similarity task (None means no normalization). $(\beta, \#\text{Iter}) = (2, 5)$ achieved the highest score. So hereafter, we applying I-C+SN+L with the hyperparameter $(\beta, \#\text{Iter}) = (2, 5)$.

Table 2: Monolingual word similarity Task; Average Spearman rank coefficient

| None | $(\beta, \#\text{Iter}) = (2, 2)$ | $(\beta, \#\text{Iter}) = (2, 3)$ | $(\beta, \#\text{Iter}) = (2, 4)$ | $(\beta, \#\text{Iter}) = (2, 5)$ |
|------|------|------|------|------|
| 0.651 | 0.67077 | 0.67101 | 0.67108 | **0.67111** |

Note that the proposed approach (I-C+SN+L) only increased this score for these similarity tasks, so showed no signs of distorting inherent information. Although Spectral Normalization does not exactly preserve the linear properties or angular properties (as centering and length normalization do, one each, respectively), it does not suffer ill effects. We hypothesize this is because it is somewhat uniformly stretching words along the major modes of variation, and is effectively removing information not relevant to meaning, like frequency of occurrence. This benign effect is on contrast to other spectral adjustments (removing small singular values, or setting all to the same value) shown in Appendix C.

## 4.2 BLI Performance across Normalization and Alignment Algorithms

We compare and evaluate the BLI performance (MAP) of various normalization algorithms from previous works to our proposed algorithms. Using the MUSE translation dictionary, we trained CCA, PROC, PROC-B and RCSLS on 5k source words and evaluated on 1.5k source test queries. The following normalization algorithms were used in the comparison analysis: PR (PCA Removal) [31], GeoMediaN (Geometric Median Normalization), C+L (Mean centering and Length normalization, 1 round), I-C+L (Iterative Mean centering and Length normalization, 5 rounds) [47], SN (Spectral Normalization, 1 round), C+SN+L (Mean centering, Spectral Normalization and Length normalization, 1 round), and I-C+SN+L (Iterative Mean centering, Spectral Normalization and Length normalization, 5 rounds). Specifically, we evaluated 18 language pairs, i.e., English (EN) from/to Bulgarian (BG), Catalan (CA), Czech (CS), German (DE), Spanish (ES), Korean (KO), Thai (TH) and Chinese (ZH), separate from hyperparameter tuning. The average is reported in Table 3, all results are in Appendix G. For almost all algorithms I-C+SN+L achieves the best scores (and especially on $\mathbf{X}_{L_2}-\mathbf{EN}$, often considerably better). The only exceptions are on non-rigid RCSLS when C+SN+L (with no iteration) or just SN (with C+L) performs slightly better. So, Spectral Normalization, and in particular I-C+SN+L, is shown as the best way to normalize languages before alignment.

Table 3: BLI performance (MAP) on aligning $\mathbf{EN}-\mathbf{X}_{L_2}$ and $\mathbf{X}_{L_2}-\mathbf{EN}$

| Normalization | Methods : $\mathbf{EN}-\mathbf{X}_{L_2}$ | | | | Methods : $\mathbf{X}_{L_1}-\mathbf{EN}$ | | | |
|---|---|---|---|---|---|---|---|---|
| | CCA | PROC | PROC-B | RCSLS | CCA | PROC | PROC-B | RCSLS |
| None | 0.358 | 0.365 | 0.377 | 0.394 | 0.398 | 0.399 | 0.405 | 0.428 |
| PR | 0.394 | 0.391 | 0.404 | 0.373 | 0.434 | 0.430 | 0.442 | 0.425 |
| GeoMediaN | 0.393 | 0.391 | 0.400 | 0.379 | 0.433 | 0.432 | 0.440 | 0.429 |
| C+L | 0.393 | 0.394 | 0.408 | 0.404 | 0.439 | 0.437 | 0.445 | 0.464 |
| I-C+L | 0.394 | 0.395 | 0.410 | 0.406 | 0.439 | 0.438 | 0.448 | 0.460 |
| SN | 0.391 | 0.394 | 0.408 | 0.405 | 0.440 | 0.438 | 0.451 | **0.468** |
| C+SN+L | 0.395 | 0.396 | 0.413 | **0.407** | 0.444 | 0.444 | 0.458 | 0.466 |
| I-C+SN+L | **0.396** | **0.398** | **0.414** | 0.406 | **0.445** | **0.446** | **0.461** | 0.466 |

We also compute the average BLI MAP score across all 28 language pair for more direct comparison to prior work [15], summarized in Table 4 and Appendix D. All results are in Appendix H. We compare I-C+SN+L (denoted with SN) against no normalization on various dictionary sizes: 1k, 3k and 5k source words and evaluated on 2k source test queries. In all cases, I-C+SN+L significantly improves over the baseline. This includes improvement over RCSLS which is non-rigid, so in principle could "learn" adjustments similar to our normalization in the process of alignment. We also tested on VECMAP, an unsupervised approach; I-C+SN+L preprocessing also improves this result from 0.375 to 0.410.

Table 4: Summary of BLI performance (MAP), average scores for all 28 language pairs. No normalization results from [15], against I-C+SN+L (denoted SN).

| Dict | CCA | CCA$^{SN}$ | PROC | PROC$^{SN}$ | PROC-B | PROC-B$^{SN}$ | DLV | DLV$^{SN}$ | RCSLS | RCSLS$^{SN}$ |
|---|---|---|---|---|---|---|---|---|---|---|
| 1K | .289 | **.314** | .299 | **.326** | .379 | **.407** | .289 | **.332** | **.331** | **.331** |
| 3K | .378 | **.401** | .384 | **.408** | .398 | **.415** | .381 | **.429** | .415 | **.427** |
| 5K | .400 | **.423** | .405 | **.429** | – | – | .403 | **.452** | .437 | **.460** |

## 4.3 Downstream Tasks

We conclude by demonstrating that Spectral Normalization not only improves in direct translation tasks, but also captures an important global structure that generalizes from a high resource language (i.e., English, EN) to lower resource languages. In both examples, a powerful classifier is trained on the EN embedding (after normalization), and then we demonstrate that after a lower resource language (e.g., German, DE) has been normalized and align the analysis task can be directly applied to that language. And in particular, adding the simple process of our normalization (I-C+SN+L) dramatically improves the results over not doing that step.

**Cross-lingual Document Classification (CLDC).** The CLDC task builds a topic classification using a language model on a high resource language (in our case English EN) across 15 topics. The TED CLDC corpus assembled by [19] was used for training and evaluation. Following [15] a simple CNN was used to train. Table 5 summarizes the average F1-score for all topic classifiers on 5 language pairs. The CLWEs induced by PROC$^{SN}$, PROC-B$^{SN}$, DLV$^{SN}$, and RCSLS$^{SN}$ (using I-C+SN+L) outperformed the baseline result (with no normalization) on the CLDC task, significantly improving the best average score from $0.421$ to $0.461$. Glavas *et.al.* [15] use only 12 of 15 topics, but could not confirm which, so we re-ran all baselines using all 15 topics.

Table 5: CLDC performance (micro-averaged $F_1$ scores). Cross-lingual transfer EN–X

| Model | Dict | EN-DE | EN-FR | EN-IT | EN-RU | EN-TR | Avg |
|---|---|---|---|---|---|---|---|
| PROC | 5K | .366 | .258 | .338 | .288 | .278 | .306 |
| PROC$^{SN}$ | 5K | .436 | .366 | .427 | .517 | .511 | **.451** |
| PROC-B | 3K | .364 | .304 | .299 | .336 | .317 | .324 |
| PROC-B$^{SN}$ | 3K | .448 | .396 | .423 | .522 | .517 | **.461** |
| DLV | 5K | .419 | .336 | .397 | .493 | .458 | .421 |
| DLV$^{SN}$ | 5K | .433 | .323 | .406 | .499 | .472 | **.427** |
| RCSLS | 5K | .466 | .397 | .403 | .403 | .406 | .415 |
| RCSLS$^{SN}$ | 5K | .468 | .500 | .443 | .488 | .394 | **.459** |

**Cross-lingual Natural Language Inference (XNLI).** We evaluated the CLWE on a cross-lingual natural language inference (XNLI) task. We used a multi-lingual XNLI corpus created by [9], which is a collection of sentence pairs from the English MultiNLI corpus [44] translated into 14 languages. The MultiNLI corpus contains 433k sentence pairs with the labels entailment, contradiction, and neutral. The intersection between XNLI languages and BLI languages result in four XNLI evaluation pairs: EN-DE, EN-FR. EN-TR and EN-RU. We use the training setup in [15] with the Enhanced Sequential Inference Model [7] on English after normalization. First, we aligned normalized versions of each language onto the normalized EN embedding to obtain the shared cross-lingual embedding. Then we used the 5k test pairs from the XNLI corpus to evaluate each language alignment. Table 6 shows the result for PROC, PROC-B, and RCSLS alignments (DLV and VECMAP transform the EN embedding in the process, so were omitted). We compare against the same procedure *without*

normalization from Glavas *et.al.* [15] (I-C+SN+L normalization denoted SN). As in other experiments, our normalization improves the average test accuracy with each alignment approach.

Table 6: XNLI performance (test set accuracy)

| Model | Dict | EN-DE | EN-FR | EN-TR | EN-RU | Avg |
|-------|------|-------|-------|-------|-------|-----|
| PROC | 5K | .607 | .534 | .568 | .585 | .574 |
| PROC$^{SN}$ | 5K | .611 | .638 | .542 | .596 | **.597** |
| PROC-B | 3K | .615 | .532 | .573 | .599 | .580 |
| PROC-B$^{SN}$ | 3K | .624 | .638 | .548 | .601 | **.603** |
| RCSLS | 5K | .390 | .363 | .387 | .399 | .385 |
| RCSLS$^{SN}$ | 5K | .499 | .482 | .504 | .556 | **.510** |

## 5  Conclusion & Discussion

We introduce a new way to normalize embeddings, based on spectral normalization, for use in creating cross-lingual word embeddings. Our approach generalizes previous approaches, and effectively removes much of the clustering of words based on properties other than the similarity which encodes meaning. When used to individually preprocess monolingual embeddings, our approach allows any alignment procedure to find better alignments: resulting in improved performance on direct translation tasks as well as cross-lingual topic classification and natural language inference tasks.

**Social impacts.**  The vast majority of NLP research and cutting-edge advancements are in English. This disadvantages those who primarily operate in other languages, with less developed models, or less data to train models. As large language models are the cornerstone of most NLP research and development in English, one of this work's main goals is to port these advances to other languages, and those who use them. This will help unlock this technology to many others around the world. As with most models, this accuracy and improvement may vary across tasks and languages.

While language models have many positive use cases including improving accessibility, better recommendations, and increased automation, they have some negative effects as well. These include requiring potentially large computational and hence environmental cost, encoding and exacerbating bias, and aiding in automatically generating fake or deceitful content. While this paper is unlikely to change the *desire* to use embeddings, it aims to reduce the burden of use and increase the effectiveness in lower-resource settings. And in particular to port models trained in English to other languages. This would reduce the cost of retraining in other languages if the English model can be reused, easing environmental costs. We support the maturing efforts in attenuating bias in all such embeddings. And while we acknowledge the possibility of this work aiding in the automated creation of deceitful content and the harm it can cause, we believe the many benefits outweigh the harms.

**Limitations.**  The overarching goal in this line of work is to port the many advances built on embeddings from high-resource settings (like built on the English language), to lower-resource settings (like Turkish). This work can apply a powerful model built on an English language model (e.g., for natural language inference) and automatically invoke it in Turkish after the embeddings have been aligned. However, the alignment will not be any better than the low-resource embedding. If the embedding is too noisy or limited, then the analysis will likely not be effective.

Also, this work focuses on non-contextual embeddings like FastText, but not contextual ones like RoBERTa which have proven almost universally more effective in NLP on English. While in principle a normalization function and alignment could be *applied* to contextual settings, we are unaware of any technique for *learning* these mappings. We believe it could be important future work.

**Data, Code, and Experiments.**  All existing methods are compared with publicly available code with publicly available data, with links above or in references. The exception is code for CLDC and XNLI is shared by Glavás *et.al.* [15]. Everything is run with default parameters; the exception is RCSLS were we follow the suggested hyperparameter selection strategy [21] (with learning rate in $\{1, 10, 25, 50\}$ and epoch number in $\{10, 20\}$). Our new code for SpecNorm is in Appendix E.

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
