**Appendix**

 **A   Weiszfeld Algorithm**

540 The Weiszfeld algorithm is an iterative method for finding the geometric median of a set of points in
541 Euclidean space based on the reformulation of a stationary point that satisfies $\nabla f(\mathbf{x}) = 0$.

542 If iteration function $T : \mathbb{R}^d \to \mathbb{R}^d$ is defined by:

$$T(\mathbf{x}) = \begin{cases} \widetilde{T}(\mathbf{x}) = \dfrac{\sum\limits_{i=1}^{n} \|\mathbf{a}_i - \mathbf{x}\|^{-1} \mathbf{a}_i}{\sum\limits_{i=1}^{n} \|\mathbf{a}_i - \mathbf{x}\|^{-1}} & \text{if} \quad \mathbf{x} \quad \notin \quad \{\mathbf{a}_1, \dots \mathbf{a}_n\} \\[3em] \mathbf{a}_i & \text{if} \quad \mathbf{x} \quad = \quad \mathbf{a}_i \ , \ i = 1, \dots, n \end{cases} \tag{1}$$

543 then the Weiszfeld algorithm is:

$$\mathbf{x}_{k+1} = T(\mathbf{x}_k), \ k \in \mathbb{N} \tag{2}$$

544 where $\mathbf{x}_0 \in \mathbb{R}^d$ is a starting point. When the current iterate, $\mathbf{x}_k \notin \{\mathbf{a}_1, \dots \mathbf{a}_n\}$, $T(\mathbf{x}_k) = \widetilde{T}(\mathbf{x}_k)$; else,
545 if $\mathbf{x}_k = \mathbf{a}_i$, then $T(\mathbf{x}_k) = \mathbf{a}_i$.

546 The Weiszfeld algorithm is presented in Algorithm 4 below:

---

**Algorithm 4** Weiszfeld algorithm (WA)

---

**Input**: Anchor points, $(\mathbf{a}_1, \dots \mathbf{a}_n)$, $\mathbf{x}_0 \in \mathbb{R}^d$ and $\epsilon > 0$
1: $k \leftarrow 0$
2: **while** True **do**
3:     $\mathbf{x}_{k+1} \leftarrow T(\mathbf{x}_k)$
4:     **if** $\|\mathbf{x}_{k+1} - \mathbf{x}_k\|_2 < \epsilon$ **then**
5:         **return** $\mathbf{x}_{k+1}$
6:     $k \leftarrow k + 1$

---

547 **A.1   Property of Geometric Median A.1**

548 **Theorem A.1.** *If* $\mathbf{x} \in \mathbb{R}^d$ *is distinct from all the given anchor points,* $\mathbf{a}_i$*, then* $\mathbf{x}$ *is the geometric*
549 *median* $\Longleftrightarrow$

$$\mathbf{0} = \sum_{i=1}^{n} \frac{\mathbf{a}_i - \mathbf{x}}{\|\mathbf{a}_i - \mathbf{x}\|} \tag{3}$$

550 *Proof.* $(\Rightarrow)$ Suppose $\mathbf{x} \in \mathbb{R}^d$ is distinct from all the given anchor points, $\mathbf{a}_i$ and $\mathbf{x}$ is the geometric
551 median of Eq. (3.1) such that $\mathbf{x} = \widetilde{T}(\mathbf{x})$ then

$$\nabla f(\mathbf{x}) = \sum_{i=1}^{n} \frac{\mathbf{a}_i - \mathbf{x}}{\|\mathbf{a}_i - \mathbf{x}\|} = 0$$

552 $(\Leftarrow)$ Suppose $\mathbf{x} \in \mathbb{R}^d$ is distinct from all the given anchor points, $\mathbf{a}_i$ and $\mathbf{x}$ is the unique optimal
553 solution of Eq. (3.1) such that $\nabla f(\mathbf{x}) = 0$ then solving for $\mathbf{x}$ while ignoring the dependency $\mathbf{x}$ in
554 $\|\mathbf{a}_i - \mathbf{x}\|$ yields:

$$\mathbf{x} = \frac{\sum\limits_{i=1}^{n} \|\mathbf{a}_i - \mathbf{x}\|^{-1} \mathbf{a}_i}{\sum\limits_{i=1}^{n} \|\mathbf{a}_i - \mathbf{x}\|^{-1}}$$

555 which is the geometric median. $\qquad\square$

## B  Spectral Statistics and Spectral Isomorphism Measures

We also explored other spectral statistics on monolingual embeddnigs. The *numeric rank* of $A$ is a smoother analog to rank (where there is noise in low rank components), defined $\eta(A) = \|A\|_F^2/\|A\|_2^2$. The *condition number* of $A$ is $\kappa(A) = \sigma_1/\sigma_d$, and measures how close the matrix is to being truly full rank, smaller is more stable. For two matrices $A_1$ and $A_2$, the *condition number harmonic mean* is $\text{COND-HM}(A_1, A_2) = \frac{2\kappa(A_1)\kappa(A_2)}{\kappa(A_1)+\kappa(A_2)}$. Smaller means the matrices are more comparable. Figure 2 plots these measures, and again demonstrates that I-C+SN+L improves these measures on matrices.

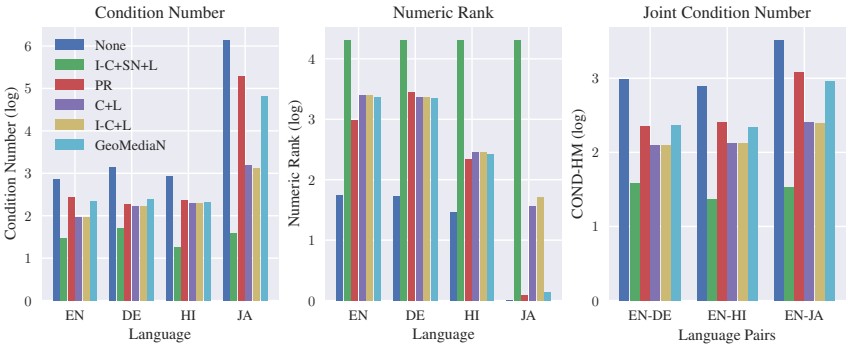

Figure 2: Spectral Measures of four (4) monolingual word embeddings.

We also show the raw numbers used to generate the charts in Figure 1 in the tables below.

Table 7: Effective Rank

|  | Normalization Algorithms | | | | | |
| --- | --- | --- | --- | --- | --- | --- |
| Langauges | None | PR | GeoMediaN | C+L | I-C+L | I-C+SN+L |
| EN | 268 | 277 | 278 | 279 | 279 | 283 |
| DE | 264 | 273 | 273 | 274 | 274 | 278 |
| HI | 258 | 270 | 269 | 269 | 269 | 282 |
| JA | 39 | 96 | 171 | 253 | 255 | 276 |

Table 8: Effective Condition Number

|  | Normalization Algorithms | | | | | |
| --- | --- | --- | --- | --- | --- | --- |
| Langauges | None | PR | GeoMediaN | C+L | I-C+L | I-C+SN+L |
| EN | 13.1 | 6.7 | 5.4 | 5.3 | 5.3 | 3.3 |
| DE | 14.8 | 5.9 | 6.2 | 6.1 | 6.1 | 3.7 |
| HI | 16.1 | 9.4 | 9.1 | 8.9 | 8.9 | 3.2 |
| JA | 175.0 | 113.3 | 73.1 | 16.4 | 15.0 | 3.4 |

Table 9: Effective Condition Number Harmonic Mean

| Langauges Pairs | Normalization Algorithms | | | | | |
|---|---|---|---|---|---|---|
| | None | PR | GeoMediaN | C+L | I-C+L | I-C+SN+L |
| EN-DE | 13.9 | 6.3 | 5.8 | 5.7 | 5.7 | 3.5 |
| EN-HI | 14.4 | 7.8 | 6.8 | 6.6 | 6.6 | 3.2 |
| EN-JA | 24.4 | 12.7 | 10.0 | 8.0 | 7.8 | 3.3 |

Table 10: Singular Value Gap

| Langauges Pairs | Normalization Algorithms | | | | | |
|---|---|---|---|---|---|---|
| | None | PR | GeoMediaN | C+L | I-C+L | I-C+SN+L |
| EN-DE | 2.3 | 2.4 | 2.2 | 2.2 | 2.2 | 2.2 |
| EN-HI | 49.0 | 48.6 | 12.2 | 12.2 | 12.2 | 5.2 |
| EN-JA | 44.1 | 45.8 | 404.0 | 17.7 | 14.8 | 2.2 |

## C  Word Similarity Task

The word similiarity task was conducted using the following English word similarity benchmarks: WS-3533 [14], WS-SIM and WSREL [1], RG-65 [35], MC-30 [30], MTurk-2875 [34], MTurk-771 [18], MEN7 [5], YP-130 [46], Rare Words [26].

In addition to a baseline (None, which means no normalization), Tabel 11 shows results comparing against our proposed normalization (I-C+SN+L) and the state of the art (I-C+L). Note that both of these improves the accuracy of the similarity tests. This indicates that they are not distorting the critical information contained in the original embeddings. And our proposed approach (I-C+SN+L) increases the score the most.

In contrast, we show the results of two other spectral adjustments we considered. SSV (SSV means Same Singular Values), performs the SVD on the original embedding and then sets all the singular values to $\eta = \sqrt{\|A\|_F^2/d}$. Then we compute U@S@VT to get the new word embedding. So similar to SpecNorm, but it never reaches the else condition. This slightly decreases the scores on the similarity tests. This is a result of accentuating the noise directions. In the other direction, SSVZ (SSVZ means Set Singular Values to Zero) only keeps the top 40 singular values and then set the rest to zero. Then we compute U@S@VT to get the new word embedding. This drastically reduces the similarity score. This shows those noise directions, which tend to be well below the average singular value, are still important and cannot just be removed.

Table 11: Monolingual Word Similarity Score (Spearman rank coefficient)

| | Normalization Algorithms | | | | |
|---|---|---|---|---|---|
| Dataset | None | SSV | SSVZ | I-C+L | I-C+SN+L |
| EN_WS-353-ALL | 0.7388 | 0.7127 | 0.5395 | 0.7433 | 0.7555 |
| EN_VERB-143 | 0.3973 | 0.4283 | 0.2635 | 0.4231 | 0.4346 |
| EN_YP-130 | 0.5333 | 0.5534 | 0.3904 | 0.5514 | 0.5631 |
| EN_MTurk-771 | 0.6689 | 0.6583 | 0.5540 | 0.6838 | 0.6926 |
| EN_RG-65 | 0.7974 | 0.7640 | 0.6390 | 0.8082 | 0.8087 |
| EN_RW-STANFORD | 0.5080 | 0.5569 | 0.3873 | 0.5125 | 0.5258 |
| EN_SEMEVAL17 | 0.7216 | 0.7478 | 0.5779 | 0.7288 | 0.7366 |
| EN_MEN-TR-3k | 0.7637 | 0.7506 | 0.6581 | 0.7720 | 0.7792 |
| EN_WS-353-SIM | 0.7811 | 0.7678 | 0.6162 | 0.7897 | 0.7888 |
| EN_MTurk-287 | 0.6773 | 0.6439 | 0.6016 | 0.6864 | 0.6864 |
| EN_WS-353-REL | 0.6820 | 0.6363 | 0.4824 | 0.6905 | 0.7081 |
| EN_MC-30 | 0.8123 | 0.8203 | 0.6754 | 0.8352 | 0.8494 |
| EN_SIMLEX-999 | 0.3823 | 0.4069 | 0.2276 | 0.3899 | 0.3955 |
| Avg | 0.6511 | 0.6498 | 0.5087 | 0.6627 | **0.6711** |

## D  Summary of Model Performance

Table 12: Summary of Model Performance on I-C+SN+L (denoted $SN$)

| | Models | | | | | |
|---|---|---|---|---|---|---|
| Dict | $CCA^{SN}$ | $PROC^{SN}$ | $PROC\text{-}B^{SN}$ | $DLV^{SN}$ | $RCSLS^{SN}$ | $VECMAP^{SN}$ |
| 1K | 28/28 | 28/28 | 25/28 | 28/28 | 13/28 | |
| 3K | 28/28 | 28/28 | 25/28 | 28/28 | 25/28 | |
| 5K | 28/28 | 28/28 | | 28/28 | 28/28 | |
| – | | | | | | 27/28 |

Table 12 summarizes the performance of I-C+SN+L on several supervised and unsupervised projection-based CLWE models across all the 28 language pairs as presented in H . After preprocessing the monolingual word embeddings with I-C+SN+L, $CCA^{SN}$, $PROC^{SN}$ and $DLV^{SN}$ outperformed CCA, PROC and DLV respectively on 28 of 28 language pairs across all the translation dictionaries. $PROC\text{-}B^{SN}$ outperformed PROC-B on 25 of 28 language pairs across 1k and 3k translation dictionaries. The performance of $RCSLS^{SN}$ supersede RCSLS on 25 of 28 language pairs and 28 of 28 language pairs trained on 3k and 5k translation dictionaries respectively. The unsupervised projection-based CLWE model, $VECMAP^{SN}$ outperformed VECMAP on 27 of 28 language pairs. The lowest performing model was $RCSLS^{SN}$ trained on 1k translation dictionary.

## E  Code for SpecNorm

The code implementing our new algorithm Spectral Normalization is quite simple, as such we just present it below. We will provide public links to the code online after double blind review.

Spectral Normalization (SpecNorm) is below, referred to as **SVDConX.py**. Similarly, the code implementation of Iterative Spectral Normalization (I-C+SN+L) referred to as **Iter_SVDConX.py** is also shown below.

**598  Spectral Normalization (SVDConX.py)**

```python
import numpy as np

def computeSVD(embeds):
    """
    Args:
        embeds: Monolingual Embedding

    Returns:
        Singular Value Decompostion
    """
    U, S, VT = np.linalg.svd(embeds,full_matrices=False)
    return U, S, VT

def SVDConX(embeds, beta):
    """
    Args:
        emdeds: Monolingual Embedding
        beta: Use to determine smaller (noisy)
        singular values to be removed

    Returns:
        Spectral Normalised Embedding
    """
    # Perform SVD on the Monolingual Embedding
    _, S, VT = computeSVD(embeds)
    # Compute eta
    eta = np.sqrt(np.sum(S**2)/len(S))
    # Transform  diagonal matrix
    S_prime = 1 / S
    for idx, sigma in enumerate(S):
        if sigma > beta*eta:
            S_prime[idx] = S_prime[idx] * (beta*eta)
        else:
            S_prime[idx] = 1
    S_prime = np.eye(len(S)) * S_prime
    # Compute new monolingual embedding
    embeds = embeds @ VT.T @ S_prime
    return embeds
```

**599  Iterative Spectral Normalization (Iter_SVDConX.py)**

```python
import numpy as np
from SVDConX import SVDConX
from argparse import ArgumentParser

def load_embed(filename, max_vocab=-1):
    words, embeds = [], []
    with open(filename, 'r') as f:
        next(f)
        for line in f:
            word, vector = line.rstrip().split(' ', 1)
            vector = np.fromstring(vector, sep=' ')
            words.append(word)
            embeds.append(vector)
            if len(embeds) == max_vocab:
                break
    return words, np.array(embeds)
```

```python
def saveEmbed(path, words, word_embeds):
    with open(path, 'w') as f:
        print(word_embeds.shape[0], word_embeds.shape[1], file=f)
        for word, embed in zip(words, word_embeds):
            vector_str = ' '.join(str(x) for x in embed)
            print(word, vector_str, file = f)

def main():
    parser = ArgumentParser()
    parser.add_argument('--input_file')
    parser.add_argument('--output_file')
    parser.add_argument('--niter', default=5, type=int)
    parser.add_argument('--max_vocab', default=200000, type=int)
    parser.add_argument('--beta', default=2, type=int)
    args = parser.parse_args()

    words, embeds = load_embed(args.input_file, max_vocab=args.max_vocab)
    embeds /= np.linalg.norm(embeds, axis=1)[:, np.newaxis] + 1e-8

    for i in range(args.niter):
        # Center Monoligual Embedding
        embeds -= embeds.mean(axis=0)[np.newaxis, :]
        # Perform Spectral Normalization
        embeds =  SVDConX(embeds, args.beta)
        # Unit Length Normalization
        embeds /= np.linalg.norm(embeds, axis=1)[:, np.newaxis] + 1e-8
    saveEmbed(args.output_file, words, embeds)

if __name__ == '__main__':
    main()
```

# F   Runtime

Most of the alignment algorithms run on a CPU except for VecMap, which requires a GPU for faster computation. It takes about 91 seconds to run Iterative Spectral Normalization on a CPU with a $\beta = 2$ and five iterations. Hardware specifications are NVIDIA GeForce GTX Titan Xp 12GB, AMD Ryzen 7 1700 eight-core processor, and 62.8GB RAM. All alignment approaches completed in under 15 minutes, and most less than 5 minutes. Each evaluation (BLI, CLDC, or XNLI) takes under 2 minutes, but the training step for CLDC and XNLI takes about a day each; hence our approach aims only to need to do this once (on a high resource language like English), and then use the faster alignment step to transfer this to other languages.

# G Full BLI performance of various normalization algorithms

Table 13: BLI performance (MAP) on aligning $\mathbf{EN-X}_{L_2}$. We compare all the normalization techniques: None (No normalization), PR (PCA Removal) [31], GeoMediaN (Geometric Median Normalization), C+L (Mean centering and Length normalization, 1 round), I-C+L (Iterative Mean centering and Length normalization, 5 rounds) [47], SN (Spectral Normalization, 1 round), C+SN+L (Mean centering, Spectral Normalization and Length normalization, 1 round), I-C+SN+L (Iterative Mean centering, Spectral Normalization and Length normalization, 5 rounds).

| Method | Normalization | BG | CA | CS | DE | ES | FR | KO | TH | ZH | Avg |
|---|---|---|---|---|---|---|---|---|---|---|---|
| CCA | None | 0.298 | 0.556 | 0.364 | 0.358 | 0.514 | 0.485 | 0.242 | 0.209 | 0.198 | 0.358 |
| | PR | 0.316 | 0.583 | 0.389 | 0.374 | 0.523 | 0.492 | 0.283 | 0.224 | 0.362 | 0.394 |
| | GeoMediaN | 0.316 | 0.580 | 0.383 | 0.376 | 0.524 | 0.492 | 0.277 | 0.226 | 0.362 | 0.393 |
| | C+L | 0.326 | 0.582 | 0.387 | 0.375 | 0.521 | 0.491 | 0.267 | 0.227 | 0.359 | 0.393 |
| | I-C+L | 0.326 | 0.582 | 0.387 | 0.375 | 0.521 | 0.492 | 0.267 | 0.226 | 0.371 | 0.394 |
| | SN | 0.314 | 0.580 | 0.384 | 0.370 | 0.519 | 0.494 | 0.259 | 0.223 | 0.378 | 0.391 |
| | C+SN+L | 0.329 | 0.586 | 0.389 | 0.374 | 0.523 | 0.495 | 0.262 | 0.225 | 0.376 | 0.395 |
| | I-C+SN+L | 0.328 | 0.585 | 0.388 | 0.374 | 0.524 | 0.496 | 0.258 | 0.229 | 0.378 | **0.396** |
| PROC | None | 0.296 | 0.553 | 0.363 | 0.357 | 0.509 | 0.481 | 0.255 | 0.212 | 0.255 | 0.365 |
| | PR | 0.316 | 0.575 | 0.386 | 0.371 | 0.524 | 0.492 | 0.285 | 0.223 | 0.343 | 0.391 |
| | GeoMediaN | 0.317 | 0.578 | 0.384 | 0.376 | 0.521 | 0.491 | 0.281 | 0.225 | 0.346 | 0.391 |
| | C+L | 0.327 | 0.582 | 0.390 | 0.373 | 0.520 | 0.490 | 0.279 | 0.227 | 0.354 | 0.394 |
| | I-C+L | 0.327 | 0.582 | 0.390 | 0.372 | 0.520 | 0.490 | 0.280 | 0.228 | 0.366 | 0.395 |
| | SN | 0.319 | 0.580 | 0.384 | 0.369 | 0.520 | 0.493 | 0.277 | 0.227 | 0.378 | 0.394 |
| | C+SN+L | 0.331 | 0.586 | 0.380 | 0.374 | 0.524 | 0.495 | 0.273 | 0.227 | 0.378 | 0.396 |
| | I-C+SN+L | 0.330 | 0.586 | 0.389 | 0.375 | 0.525 | 0.495 | 0.287 | 0.224 | 0.374 | **0.398** |
| PROC-B | None | 0.326 | 0.587 | 0.400 | 0.382 | 0.528 | 0.497 | 0.236 | 0.218 | 0.221 | 0.377 |
| | PR | 0.340 | 0.605 | 0.425 | 0.395 | 0.536 | 0.505 | 0.259 | 0.227 | 0.341 | 0.404 |
| | GeoMediaN | 0.304 | 0.605 | 0.425 | 0.395 | 0.538 | 0.507 | 0.260 | 0.219 | 0.352 | 0.400 |
| | C+L | 0.354 | 0.607 | 0.423 | 0.396 | 0.536 | 0.507 | 0.257 | 0.223 | 0.366 | 0.408 |
| | I-C+L | 0.354 | 0.608 | 0.424 | 0.396 | 0.536 | 0.508 | 0.257 | 0.224 | 0.380 | 0.410 |
| | SN | 0.347 | 0.602 | 0.421 | 0.392 | 0.533 | 0.504 | 0.257 | 0.229 | 0.389 | 0.408 |
| | C+SN+L | 0.358 | 0.613 | 0.427 | 0.396 | 0.539 | 0.501 | 0.261 | 0.229 | 0.397 | 0.413 |
| | I-C+SN+L | 0.358 | 0.619 | 0.426 | 0.397 | 0.538 | 0.510 | 0.258 | 0.227 | 0.393 | **0.414** |
| RCSLS | None | 0.347 | 0.601 | 0.404 | 0.392 | 0.530 | 0.503 | 0.317 | 0.227 | 0.227 | 0.394 |
| | PR | 0.337 | 0.591 | 0.387 | 0.385 | 0.529 | 0.498 | 0.290 | 0.234 | 0.107 | 0.373 |
| | GeoMediaN | 0.337 | 0.592 | 0.391 | 0.384 | 0.530 | 0.499 | 0.284 | 0.231 | 0.167 | 0.379 |
| | C+L | 0.345 | 0.599 | 0.400 | 0.391 | 0.530 | 0.502 | 0.288 | 0.221 | 0.361 | 0.404 |
| | I-C+L | 0.346 | 0.598 | 0.400 | 0.391 | 0.530 | 0.502 | 0.288 | 0.221 | 0.382 | 0.406 |
| | SN | 0.341 | 0.597 | 0.395 | 0.394 | 0.533 | 0.504 | 0.282 | 0.217 | 0.385 | 0.405 |
| | C+SN+L | 0.348 | 0.601 | 0.403 | 0.393 | 0.533 | 0.506 | 0.285 | 0.215 | 0.377 | **0.407** |
| | I-C+SN+L | 0.348 | 0.601 | 0.401 | 0.392 | 0.533 | 0.506 | 0.280 | 0.214 | 0.376 | 0.406 |

Table 14: BLI performance (MAP) on aligning $\mathbf{X}_{L_1}-\mathbf{EN}$. We compare all the normalization techniques: None (No normalization), PR (PCA Removal) [31], GeoMediaN (Geometric Median Normalization), C+L (Mean centering and Length normalization, 1 round), I-C+L (Iterative Mean centering and Length normalization, 5 rounds) [47], SN (Spectral Normalization, 1 round), C+SN+L (Mean centering, Spectral Normalization and Length normalization, 1 round), I-C+SN+L (Iterative Mean centering, Spectral Normalization and Length normalization, 5 rounds).

| Method | Normalization | BG | CA | CS | DE | ES | FR | KO | TH | ZH | Avg |
|---|---|---|---|---|---|---|---|---|---|---|---|
| CCA | None | 0.448 | 0.673 | 0.514 | 0.444 | 0.576 | 0.568 | 0.199 | 0.086 | 0.078 | 0.398 |
| | PR | 0.465 | 0.684 | 0.523 | 0.450 | 0.581 | 0.578 | 0.230 | 0.099 | 0.292 | 0.434 |
| | GeoMediaN | 0.467 | 0.688 | 0.523 | 0.449 | 0.582 | 0.583 | 0.231 | 0.098 | 0.279 | 0.433 |
| | C+L | 0.471 | 0.692 | 0.526 | 0.449 | 0.582 | 0.585 | 0.235 | 0.102 | 0.306 | 0.439 |
| | I-C+L | 0.471 | 0.692 | 0.526 | 0.449 | 0.582 | 0.585 | 0.234 | 0.102 | 0.313 | 0.439 |
| | SN | 0.467 | 0.689 | 0.527 | 0.455 | 0.587 | 0.581 | 0.230 | 0.114 | 0.310 | 0.440 |
| | C+SN+L | 0.472 | 0.693 | 0.527 | 0.458 | 0.586 | 0.590 | 0.238 | 0.115 | 0.319 | 0.444 |
| | I-C+SN+L | 0.473 | 0.692 | 0.526 | 0.459 | 0.586 | 0.590 | 0.236 | 0.115 | 0.324 | **0.445** |
| PROC | None | 0.450 | 0.669 | 0.510 | 0.440 | 0.573 | 0.569 | 0.203 | 0.081 | 0.096 | 0.399 |
| | PR | 0.465 | 0.679 | 0.519 | 0.447 | 0.578 | 0.579 | 0.235 | 0.099 | 0.273 | 0.430 |
| | GeoMediaN | 0.468 | 0.685 | 0.519 | 0.449 | 0.581 | 0.582 | 0.236 | 0.100 | 0.267 | 0.432 |
| | C+L | 0.475 | 0.688 | 0.523 | 0.451 | 0.582 | 0.583 | 0.240 | 0.101 | 0.293 | 0.437 |
| | I-C+L | 0.475 | 0.688 | 0.523 | 0.451 | 0.582 | 0.583 | 0.240 | 0.103 | 0.301 | 0.438 |
| | SN | 0.470 | 0.687 | 0.523 | 0.452 | 0.584 | 0.580 | 0.234 | 0.116 | 0.298 | 0.438 |
| | C+SN+L | 0.475 | 0.692 | 0.526 | 0.457 | 0.586 | 0.589 | 0.245 | 0.115 | 0.315 | 0.444 |
| | I-C+SN+L | 0.476 | 0.694 | 0.527 | 0.458 | 0.586 | 0.589 | 0.245 | 0.115 | 0.321 | **0.446** |
| PROC-B | None | 0.453 | 0.675 | 0.531 | 0.458 | 0.576 | 0.579 | 0.211 | 0.077 | 0.085 | 0.405 |
| | PR | 0.477 | 0.693 | 0.546 | 0.465 | 0.585 | 0.587 | 0.253 | 0.110 | 0.261 | 0.442 |
| | GeoMediaN | 0.476 | 0.691 | 0.545 | 0.469 | 0.585 | 0.590 | 0.251 | 0.107 | 0.242 | 0.440 |
| | C+L | 0.483 | 0.698 | 0.550 | 0.468 | 0.584 | 0.590 | 0.259 | 0.111 | 0.264 | 0.445 |
| | I-C+L | 0.483 | 0.698 | 0.550 | 0.469 | 0.583 | 0.590 | 0.255 | 0.113 | 0.290 | 0.448 |
| | SN | 0.479 | 0.697 | 0.553 | 0.470 | 0.588 | 0.590 | 0.251 | 0.133 | 0.302 | 0.451 |
| | C+SN+L | 0.489 | 0.702 | 0.555 | 0.474 | 0.591 | 0.598 | 0.261 | 0.127 | 0.325 | 0.458 |
| | I-C+SN+L | 0.491 | 0.703 | 0.558 | 0.475 | 0.592 | 0.599 | 0.258 | 0.130 | 0.341 | **0.461** |
| RCSLS | None | 0.509 | 0.721 | 0.556 | 0.463 | 0.612 | 0.607 | 0.265 | 0.120 | 0.003 | 0.428 |
| | PR | 0.505 | 0.724 | 0.548 | 0.464 | 0.611 | 0.611 | 0.249 | 0.077 | 0.035 | 0.425 |
| | GeoMediaN | 0.504 | 0.725 | 0.549 | 0.462 | 0.611 | 0.611 | 0.250 | 0.108 | 0.041 | 0.429 |
| | C+L | 0.510 | 0.728 | 0.549 | 0.462 | 0.612 | 0.613 | 0.259 | 0.116 | 0.327 | 0.464 |
| | I-C+L | 0.510 | 0.728 | 0.549 | 0.463 | 0.612 | 0.613 | 0.260 | 0.118 | 0.285 | 0.460 |
| | SN | 0.510 | 0.732 | 0.553 | 0.467 | 0.613 | 0.615 | 0.253 | 0.119 | 0.349 | **0.468** |
| | C+SN+L | 0.505 | 0.729 | 0.549 | 0.466 | 0.612 | 0.610 | 0.251 | 0.118 | 0.354 | 0.466 |
| | I-C+SN+L | 0.505 | 0.727 | 0.549 | 0.466 | 0.612 | 0.610 | 0.251 | 0.118 | 0.352 | 0.466 |

## H  Full BLI results for all 28 language pairs, translation dictionaries, and models.

Table 15: BLI performance (MAP) for the first batch (14) of language pairs. We compared the Baseline result from [15] to I-C+SN+L (denoted SN) result on the BLI task.

| Model | Dict | DE-FI | DE-FR | DE-HR | DE-IT | DE-RU | DE-TR | EN-DE | EN-FI | EN-FR | EN-HR | EN-IT | EN-RU | EN-TR | FI-FR | Avg |
|---|---|---|---|---|---|---|---|---|---|---|---|---|---|---|---|---|
| CCA | 1K | 0.241 | 0.422 | 0.206 | 0.414 | 0.308 | 0.153 | 0.458 | 0.259 | 0.582 | 0.218 | 0.538 | 0.336 | 0.218 | 0.230 | 0.327 |
| CCA$^{SN}$ | 1K | 0.259 | 0.456 | 0.224 | 0.445 | 0.326 | 0.179 | 0.486 | 0.286 | 0.609 | 0.244 | 0.560 | 0.362 | 0.259 | 0.260 | **0.354** |
| CCA | 3K | 0.328 | 0.494 | 0.298 | 0.491 | 0.399 | 0.251 | 0.531 | 0.351 | 0.642 | 0.299 | 0.613 | 0.434 | 0.314 | 0.332 | 0.413 |
| CCA$^{SN}$ | 3K | 0.345 | 0.518 | 0.314 | 0.511 | 0.413 | 0.278 | 0.554 | 0.377 | 0.657 | 0.335 | 0.634 | 0.455 | 0.348 | 0.360 | **0.436** |
| CCA | 5K | 0.353 | 0.509 | 0.318 | 0.506 | 0.411 | 0.280 | 0.542 | 0.383 | 0.652 | 0.325 | 0.624 | 0.454 | 0.327 | 0.362 | 0.432 |
| CCA$^{SN}$ | 5K | 0.371 | 0.528 | 0.340 | 0.527 | 0.426 | 0.303 | 0.568 | 0.410 | 0.665 | 0.356 | 0.648 | 0.476 | 0.372 | 0.387 | **0.455** |
| PROC | 1K | 0.264 | 0.428 | 0.225 | 0.421 | 0.323 | 0.169 | 0.458 | 0.271 | 0.579 | 0.225 | 0.535 | 0.352 | 0.225 | 0.239 | 0.336 |
| PROC$^{SN}$ | 1K | 0.280 | 0.459 | 0.244 | 0.458 | 0.346 | 0.194 | 0.490 | 0.293 | 0.611 | 0.255 | 0.566 | 0.378 | 0.263 | 0.268 | **0.365** |
| PROC | 3K | 0.340 | 0.499 | 0.308 | 0.495 | 0.413 | 0.260 | 0.532 | 0.365 | 0.642 | 0.307 | 0.611 | 0.449 | 0.320 | 0.333 | 0.420 |
| PROC$^{SN}$ | 3K | 0.354 | 0.522 | 0.326 | 0.516 | 0.423 | 0.283 | 0.558 | 0.385 | 0.659 | 0.346 | 0.637 | 0.472 | 0.357 | 0.362 | **0.443** |
| PROC | 5K | 0.359 | 0.511 | 0.329 | 0.510 | 0.425 | 0.284 | 0.544 | 0.396 | 0.654 | 0.336 | 0.625 | 0.464 | 0.335 | 0.362 | 0.438 |
| PROC$^{SN}$ | 5K | 0.378 | 0.531 | 0.350 | 0.531 | 0.440 | 0.312 | 0.570 | 0.421 | 0.670 | 0.366 | 0.650 | 0.490 | 0.380 | 0.388 | **0.463** |
| PROC-B | 1K | 0.354 | 0.511 | 0.306 | 0.507 | 0.392 | 0.250 | 0.521 | 0.360 | 0.633 | 0.296 | 0.605 | 0.419 | 0.301 | 0.329 | 0.413 |
| PROC-B$^{SN}$ | 1K | 0.347 | 0.531 | 0.321 | 0.518 | 0.359 | 0.283 | 0.543 | 0.411 | 0.66 | 0.346 | 0.628 | 0.414 | 0.354 | 0.373 | **0.435** |
| PROC-B | 3K | 0.362 | 0.514 | 0.324 | 0.508 | 0.413 | 0.278 | 0.532 | 0.380 | 0.642 | 0.336 | 0.612 | 0.449 | 0.328 | 0.350 | 0.431 |
| PROC-B$^{SN}$ | 3K | 0.359 | 0.535 | 0.342 | 0.524 | 0.378 | 0.293 | 0.545 | 0.415 | 0.657 | 0.362 | 0.631 | 0.443 | 0.368 | 0.376 | **0.445** |
| DLV | 1K | 0.259 | 0.384 | 0.222 | 0.420 | 0.325 | 0.167 | 0.454 | 0.271 | 0.546 | 0.225 | 0.537 | 0.353 | 0.221 | 0.209 | 0.328 |
| DLV$^{SN}$ | 1K | 0.260 | 0.472 | 0.239 | 0.458 | 0.333 | 0.198 | 0.503 | 0.305 | 0.632 | 0.274 | 0.584 | 0.389 | 0.287 | 0.274 | **0.372** |
| DLV | 3K | 0.341 | 0.496 | 0.306 | 0.494 | 0.411 | 0.261 | 0.533 | 0.365 | 0.636 | 0.307 | 0.611 | 0.444 | 0.320 | 0.321 | 0.418 |
| DLV$^{SN}$ | 3K | 0.361 | 0.540 | 0.339 | 0.537 | 0.414 | 0.300 | 0.571 | 0.418 | 0.677 | 0.381 | 0.651 | 0.471 | 0.393 | 0.399 | **0.461** |
| DLV | 5K | 0.357 | 0.506 | 0.328 | 0.510 | 0.423 | 0.284 | 0.545 | 0.396 | 0.649 | 0.334 | 0.625 | 0.467 | 0.335 | 0.351 | 0.436 |
| DLV$^{SN}$ | 5K | 0.384 | 0.549 | 0.365 | 0.548 | 0.424 | 0.326 | 0.582 | 0.449 | 0.684 | 0.404 | 0.661 | 0.488 | 0.407 | 0.431 | **0.479** |
| RCSLS | 1K | 0.288 | 0.459 | 0.262 | 0.453 | 0.361 | 0.201 | 0.501 | 0.306 | 0.612 | 0.267 | 0.565 | 0.401 | 0.275 | 0.269 | 0.373 |
| RCSLS$^{SN}$ | 1K | 0.282 | 0.465 | 0.247 | 0.459 | 0.347 | 0.197 | 0.508 | 0.305 | 0.635 | 0.266 | 0.577 | 0.403 | 0.273 | 0.271 | **0.374** |
| RCSLS | 3K | 0.373 | 0.524 | 0.337 | 0.518 | 0.442 | 0.296 | 0.568 | 0.404 | 0.665 | 0.357 | 0.637 | 0.491 | 0.364 | 0.367 | 0.453 |
| RCSLS$^{SN}$ | 3K | 0.366 | 0.543 | 0.336 | 0.533 | 0.448 | 0.302 | 0.612 | 0.421 | 0.696 | 0.375 | 0.668 | 0.523 | 0.395 | 0.374 | **0.471** |
| RCSLS | 5K | 0.395 | 0.536 | 0.359 | 0.529 | 0.458 | 0.324 | 0.580 | 0.438 | 0.675 | 0.375 | 0.652 | 0.510 | 0.386 | 0.395 | 0.472 |
| RCSLS$^{SN}$ | 5K | 0.404 | 0.569 | 0.370 | 0.550 | 0.480 | 0.345 | 0.636 | 0.465 | 0.713 | 0.419 | 0.687 | 0.557 | 0.439 | 0.416 | **0.504** |
| VECMAP | - | 0.302 | 0.505 | 0.300 | 0.493 | 0.322 | 0.253 | 0.521 | 0.292 | 0.626 | 0.268 | 0.600 | 0.323 | 0.288 | 0.368 | 0.390 |
| VECMAP$^{SN}$ | - | 0.343 | 0.539 | 0.326 | 0.533 | 0.337 | 0.293 | 0.559 | 0.355 | 0.660 | 0.333 | 0.635 | 0.368 | 0.352 | 0.400 | **0.431** |

Table 16: BLI performance (MAP) for second batch (14) of language pairs.

| Model | Dict | FI-HR | FI-IT | FI-RU | HR-FR | HR-IT | HR-RU | IT-FR | RU-FR | RU-IT | TR-FI | TR-FR | TR-HR | TR-IT | TR-RU | Avg |
|---|---|---|---|---|---|---|---|---|---|---|---|---|---|---|---|---|
| CCA | 1K | 0.167 | 0.232 | 0.214 | 0.238 | 0.240 | 0.256 | 0.612 | 0.344 | 0.352 | 0.151 | 0.213 | 0.134 | 0.202 | 0.146 | 0.250 |
| CCA$^{SN}$ | 1K | 0.193 | 0.257 | 0.236 | 0.273 | 0.265 | 0.274 | 0.638 | 0.380 | 0.379 | 0.157 | 0.236 | 0.148 | 0.227 | 0.164 | **0.273** |
| CCA | 3K | 0.264 | 0.328 | 0.306 | 0.346 | 0.345 | 0.348 | 0.659 | 0.452 | 0.449 | 0.232 | 0.308 | 0.211 | 0.309 | 0.252 | 0.343 |
| CCA$^{SN}$ | 3K | 0.289 | 0.359 | 0.331 | 0.375 | 0.377 | 0.366 | 0.672 | 0.476 | 0.469 | 0.257 | 0.332 | 0.240 | 0.329 | 0.269 | **0.367** |
| CCA | 5K | 0.288 | 0.353 | 0.340 | 0.372 | 0.366 | 0.367 | 0.668 | 0.469 | 0.474 | 0.260 | 0.337 | 0.250 | 0.331 | 0.285 | 0.369 |
| CCA$^{SN}$ | 5K | 0.311 | 0.384 | 0.362 | 0.403 | 0.393 | 0.389 | 0.681 | 0.491 | 0.492 | 0.284 | 0.364 | 0.269 | 0.357 | 0.299 | **0.391** |
| PROC | 1K | 0.187 | 0.247 | 0.233 | 0.248 | 0.247 | 0.269 | 0.615 | 0.352 | 0.360 | 0.169 | 0.215 | 0.148 | 0.211 | 0.168 | 0.262 |
| PROC$^{SN}$ | 1K | 0.217 | 0.271 | 0.252 | 0.285 | 0.276 | 0.285 | 0.641 | 0.387 | 0.391 | 0.178 | 0.243 | 0.166 | 0.239 | 0.182 | **0.287** |
| PROC | 3K | 0.269 | 0.328 | 0.310 | 0.346 | 0.350 | 0.353 | 0.659 | 0.455 | 0.455 | 0.241 | 0.312 | 0.219 | 0.312 | 0.262 | 0.348 |
| PROC$^{SN}$ | 3K | 0.296 | 0.365 | 0.337 | 0.381 | 0.384 | 0.371 | 0.671 | 0.474 | 0.472 | 0.262 | 0.336 | 0.248 | 0.336 | 0.279 | **0.372** |
| PROC | 5K | 0.294 | 0.355 | 0.342 | 0.374 | 0.364 | 0.372 | 0.669 | 0.470 | 0.474 | 0.269 | 0.338 | 0.259 | 0.335 | 0.290 | 0.372 |
| PROC$^{SN}$ | 5K | 0.316 | 0.385 | 0.364 | 0.407 | 0.396 | 0.393 | 0.679 | 0.491 | 0.495 | 0.290 | 0.368 | 0.275 | 0.360 | 0.305 | **0.395** |
| PROC-B | 1K | 0.263 | 0.328 | 0.315 | 0.335 | 0.343 | 0.348 | 0.665 | 0.467 | 0.466 | 0.247 | 0.305 | 0.210 | 0.298 | 0.230 | 0.344 |
| PROC-B$^{SN}$ | 1K | 0.296 | 0.365 | 0.337 | 0.408 | 0.392 | 0.371 | 0.678 | 0.486 | 0.483 | 0.280 | 0.357 | 0.255 | 0.346 | 0.246 | **0.379** |
| PROC-B | 3K | 0.293 | 0.348 | 0.327 | 0.365 | 0.368 | 0.365 | 0.664 | 0.478 | 0.476 | 0.270 | 0.333 | 0.244 | 0.330 | 0.262 | 0.366 |
| PROC-B$^{SN}$ | 3K | 0.303 | 0.374 | 0.337 | 0.403 | 0.399 | 0.377 | 0.678 | 0.488 | 0.491 | 0.286 | 0.360 | 0.267 | 0.356 | 0.264 | **0.384** |
| DLV | 1K | 0.184 | 0.244 | 0.225 | 0.214 | 0.245 | 0.264 | 0.585 | 0.320 | 0.358 | 0.161 | 0.194 | 0.144 | 0.209 | 0.161 | 0.251 |
| DLV$^{SN}$ | 1K | 0.217 | 0.275 | 0.249 | 0.290 | 0.286 | 0.286 | 0.645 | 0.398 | 0.393 | 0.174 | 0.266 | 0.164 | 0.252 | 0.182 | **0.291** |
| DLV | 3K | 0.269 | 0.331 | 0.307 | 0.331 | 0.348 | 0.353 | 0.653 | 0.446 | 0.452 | 0.243 | 0.306 | 0.219 | 0.311 | 0.261 | 0.345 |
| DLV$^{SN}$ | 3K | 0.324 | 0.390 | 0.364 | 0.417 | 0.415 | 0.394 | 0.684 | 0.495 | 0.495 | 0.294 | 0.373 | 0.276 | 0.361 | 0.288 | **0.398** |
| DLV | 5K | 0.294 | 0.356 | 0.342 | 0.364 | 0.366 | 0.374 | 0.665 | 0.466 | 0.475 | 0.268 | 0.333 | 0.255 | 0.336 | 0.289 | 0.370 |
| DLV$^{SN}$ | 5K | 0.357 | 0.420 | 0.392 | 0.445 | 0.440 | 0.422 | 0.695 | 0.515 | 0.513 | 0.320 | 0.401 | 0.311 | 0.391 | 0.322 | **0.425** |
| RCSLS | 1K | 0.214 | 0.272 | 0.257 | 0.281 | 0.275 | 0.291 | 0.637 | 0.381 | 0.383 | 0.194 | 0.247 | 0.170 | 0.246 | 0.191 | **0.289** |
| RCSLS$^{SN}$ | 1K | 0.217 | 0.271 | 0.253 | 0.284 | 0.279 | 0.286 | 0.645 | 0.388 | 0.393 | 0.179 | 0.243 | 0.168 | 0.239 | 0.185 | 0.288 |
| RCSLS | 3K | 0.296 | 0.362 | 0.341 | 0.384 | 0.382 | 0.379 | 0.673 | 0.477 | 0.472 | 0.272 | 0.348 | 0.256 | 0.340 | 0.290 | 0.377 |
| RCSLS$^{SN}$ | 3K | 0.301 | 0.372 | 0.345 | 0.392 | 0.388 | 0.382 | 0.684 | 0.489 | 0.482 | 0.270 | 0.363 | 0.259 | 0.348 | 0.291 | **0.383** |
| RCSLS | 5K | 0.321 | 0.388 | 0.376 | 0.412 | 0.399 | 0.404 | 0.682 | 0.494 | 0.491 | 0.300 | 0.375 | 0.285 | 0.368 | 0.324 | 0.401 |
| RCSLS$^{SN}$ | 5K | 0.331 | 0.403 | 0.392 | 0.431 | 0.417 | 0.417 | 0.700 | 0.520 | 0.509 | 0.304 | 0.397 | 0.302 | 0.385 | 0.335 | **0.417** |
| VECMAP | - | 0.280 | 0.355 | 0.312 | 0.402 | 0.389 | 0.376 | 0.667 | 0.463 | 0.463 | 0.246 | 0.341 | 0.223 | 0.332 | 0.200 | 0.361 |
| VECMAP$^{SN}$ | - | 0.289 | 0.398 | 0.350 | 0.438 | 0.431 | 0.407 | 0.689 | 0.497 | 0.487 | 0.270 | 0.386 | 0.251 | 0.365 | 0.194 | **0.389** |