# OpenReview forum: "Normalization of Language Embeddings for Cross-Lingual Alignment"
_NeurIPS.cc/2021/Conference — NeurIPS 2021 Submitted_

### Official Review · Reviewer_z8YG · 2021-06-28

**Rating:** 5
**Confidence:** 4

**Summary:**

The paper presents a preprocessing technique called ’spectral normalization’ for cross-lingual word alignment and shows improvements across a range of alignment techniques and downstream tasks.

**Limitations And Societal Impact:**

The paper presents thorough empirical analyses of a normalization technique which can potentially help cross-lingual transfer of NLP models. The focus on word alignment favors morphologically simple languages, though. In §5, you discuss related limitations, but I think this section would benefit from a discussion of morphology. A related, but broader question (which I don't think you necessarily have to address, but should be mindful of), is the impact of strong normative writing systems, and the open problem of how best to approach the majority of low-resource languages that are predominantly spoken languages.

**Main Review:**

I in many ways really like the paper. The empirical evaluation is thorough, and the normalization trick is neat. I think the paper suffers from three fundamental limitations:

a) It is not entirely clear how spectral normalization relates to previous discussions of spectrality (some of the references cited, as well as the ones below). I would have liked to see this fleshed out in more detail.
b) The normalization trick is the sole contribution; feels a little ’thin’ for a full length paper.
c) Interest in alignment of word embeddings is on the decline, since word embeddings are not used much any longer. The paper would benefit by the authors’ take on why this problem remains interesting (whether for theoretical or practical reasons).

Missing references for related work discussing spectrality in the context of cross-lingual vector space alignment, e.g.:

https://www.aclweb.org/anthology/P16-2080.pdf
https://www.aclweb.org/anthology/Q18-1014.pdf
https://www.aclweb.org/anthology/P18-1072.pdf

See also Aldarmaki’s thesis: https://www.proquest.com/openview/97f58b5d99e2ed81065054a170f2dcda/1?pq-origsite=gscholar&cbl=18750&diss=y

(I’m omitting older related work, e.g., [http://www.ra.ethz.ch/cdstore/www2010/www/p751.pdf], and basic background, e.g., [https://arxiv.org/abs/math/0203240], which I also think the authors should cite.)

Note the term ’spectral normalization’ has been used before with different meanings. It may make sense to address this somewhere. See, for example, [https://www.aclweb.org/anthology/K19-1064.pdf, https://www.aclweb.org/anthology/2020.findings-emnlp.64/].

The paragraph in line 44-50 needs references.


**Time Spent Reviewing:**

1

---

### Official Review · Reviewer_sMU3 · 2021-07-15

**Rating:** 4
**Confidence:** 5

**Summary:**

The paper proposes a new normalization method for monolingual word embeddings before aligning them between different languages. The normalization is based on spectral properties of the monolingual embeddings and aims to level the contributions of those leading singular values while maintaining those smaller ones at the same time. Further inspired by previous works that centers and units the length of monolingual embeddings, the authors propose an iterative normalization that combines the spectral normalization with centering and unit length normalization. The proposed method achieves better performance in both bilingual lexical induction (BLI) tasks as well as downstream tasks like cross-lingual document classification (CLDC) and cross-lingual natural language inference (XNLI).

**Limitations And Societal Impact:**

I agree with the limitations mentioned by the authors.

**Main Review:**

Originality: The paper builds on a well-defined framework of cross-lingual word embedding alignment and focuses on the normalization step. The proposed method is quite novel compared with previous approaches as it emphasizes on the spectrum properties of the word embeddings. The related work is also adequately cited.

Quality: In general, the paper is technically sound. However, line 18 claims "The best multilingual NLP approaches typically do not jointly learn a single embedding". This should be supported by citations or other evidence since some of the SOTA multilingual methods like XLM, XLM-R and m-BERT all learn joint multilingual embeddings. About evaluation, table 4, 5 and 6 should include other strong baselines just in the same way as table 3.

Clarity: There are few clarity issues of the paper: 1) How is the proposed method "subsuming various previous approaches"? 2) What is the position of GeoMediaN? Is it a baseline or a proposed method? In section 3 it is introduced as a proposed method but in section 4 it is treated mostly like one of the baselines. 3) It is better to be more concise in scientific writing and improve the statement like line 57 "or optimizing something else".

Significance: The empirical result is clearly written, and implementation details are also included. However, the performance gain of the proposed method (I-C+SN+L) against strong baseline (I-C+L) is marginal as shown in table 3. And the same comparison is overlooked in table 4, 5 and 6. This insignificance of empirical superiority may limit the contribution of this work to the community.

**Time Spent Reviewing:**

3

---

> ### Author Response · Authors · 2021-08-10
> **significance**
>
> Thank you for the careful review.  Here are a couple points not directly addressed in the global response that we felt useful to provide an answer for:
>
>
> 1) How is the proposed method "subsuming various previous approaches"?
>
> Our method incorporates the previous iterative normalization, but also adds spectral normalization.  The Spectral normalization implicitly handles older heuristic like PCA removal and its relatives as well.  We believe this is a universal solution for how to normalize such embeddings before alignment.
>
>
> 2) What is the position of GeoMediaN? Is it a baseline or a proposed method? In section 3 it is introduced as a proposed method but in section 4 it is treated mostly like one of the baselines.
>
> The GeoMediaN approach we propose in Algorithm 1, is a simpler alternative to the iterative normalization (I-C+L) of Zhang etal.  At least in that it achieves the same stated goals.  However, for whatever reason it does not perform well.  We included this result because we found it interesting, and it provides evidence that iterative methods have some inherent advantage.  We did not compare against it in all experiments since it was clear it was not going to be the best overall method.
>
> Significance:  “However, the performance gain of the proposed method (I-C+SN+L) against strong baseline (I-C+L) is marginal as shown in table 3”
>
> Note that the improvement is larger in the best performing alignment methods.

---

### Official Review · Reviewer_qE9E · 2021-07-19

**Rating:** 4
**Confidence:** 4

**Summary:**

This paper presents a new approach to normalizing (static) word embeddings prior to learning the mapping between them to induce an improved shared cross-lingual word embedding (CLWE) space. The main idea is to improve shared cross-lingual spaces not via learning more sophisticated mapping functions or by designing a new CLWE learning paradigm, but rather to affect the properties of independently trained monolingual embeddings to increase their alignability. In other words, normalization can be seen as a pre-processing step that is orthogonal to the actual mapping/projection functions, and improving normalization can have a positive impact on the final shared space. The core contribution of the paper is therefore the spectral normalization method, built on top of the recent advances and insights from prior work, which yields gains over other normalization techniques from prior work in the intrinsic task of Bilingual Lexicon Induction (BLI) and several transfer tasks (which rely on the CLWEs).

**Ethical Concerns:**

None.

**Limitations And Societal Impact:**

The paper lists Turkish as an example of a low-resource language, but according to Joshi et al. (ACL-20)'s analysis - there are much lower-resource languages, and this should also be mentioned in the paper.

The authors state that they focused only on static CLWEs as the limitation of the current work. This is indeed a limitation, but this paper already should have ideally provided some extensions that reach towards contextualised WE learning.

**Main Review:**

UPDATE AFTER READING THE AUTHOR RESPONSE:

I would very much like to thank the authors for providing a detailed response. The response did clarify some of my questions (e.g., how the hparams were chosen), but I still think that the paper should be revised and made stronger and more impactful, e.g., 1) by running similar experiments and empirically demonstrating the usefulness of the proposed method to contextualised embeddings; 2) by comparing to some more recent models (e.g., Ormazabal et al, ACL 2021); 3) by acknowledging other recent research (especially spectral methods) on isometry/isomorphism of word embedding spaces. Further, I am still unsure about the relationship of embed-then-align projection methods and joint learning methods and I need more evidence from prior research that joint training does not work beyond BLI (i.e., I am unsure if the claims made in the response are factually correct).

I do agree with the authors that "there are numerous applications for embeddings outside of NLP (images, graphs, spatial data, merchants, biology) where there is no concept of contextual embedding.", but then the authors should also demonstrate the usefulness of their method outside of NLP, not leaving it as speculation only.

All in all, as said - while I do like the paper and its main goals overall, I still hesitate to recommend it for acceptance.


===============
Originality/novelty
===============

While the paper isolates and discusses the crucial sub-problem of WE normalization very well, P1) it is heavily based on a plethora of prior research, and therefore feels more like a consolidation of the ideas from prior research; P2) it does not cite some of the more recent research focused on the same goal - improving CLWEs by learning more isomorphic monolingual embedding spaces to facilitate the mapping; P3) it also makes some vague statements about the importance of mapping-based CLWE approaches in modern NLP. I discuss these three points/aspects P1-P3 in what follows.

Regarding P1, the core contribution of the paper is combining the previous findings and normalization methods of Artetxe et al. (ACL 2018) and Zhang et al. (ACL 2019) with recent insights of Dubossarsky et al. (EMNLP 2020) that spectral properties of monolingual WE spaces are largely correlated with the final task performance. This paper makes a step towards the synergy of these two ideas, introducing very simple methods that 'zero out singular values' in order to create spectrally more similar monolingual WEs. This then yields some gains in the standard suite of evaluations (borrowed from the work of Glavas et al. ACL-19). While this new spectral normalization method is a nice new insight, I find it too thin to justify a full-fledged paper, especially since the motivation for why exactly this flavour of spectral normalization works (and some other possible variants don't).

Regarding P2, there has been some recent research that tried to achieve the same effects as this paper, but those works are not cited nor compared against as baselines. Listing a few papers here:
- https://arxiv.org/pdf/2012.15715.pdf -> this paper also aims to impact the monolingual WE structure to learn better mappings (via context anchoring)
- https://aclanthology.org/2020.repl4nlp-1.7.pdf -> the motivation of this very relevant paper is exactly the same as the motivation in this paper.
- https://direct.mit.edu/coli/article/46/2/257/93364/Unsupervised-Word-Translation-with-Adversarial -> this paper also aims to bypass the problems with non-isomorphic spaces through 'translating' the problem to a latent space (induced by autoencoders), other work from the same authors also follows a similar line of thinking.

Regarding P3, unfortunately the work stays at the level of static CLWEs without providing a proper justification nor discussion if these methods can advance contextual word embeddings which are now predominant in modern-day NLP. What is more, can the insights on normalization help us in other tasks? The current evaluation tasks (except for BLI) have been superseded by much more powerful transfer methods, while the gains in BLI (as the task where static CLWEs still hit near-SotA performance) are still very modest, which questions the usefulness of the proposed normalization technique.

Further, the authors also argue that joint learning of CLWEs cannot match mapping-based approaches. This is not in true in general -> mapping-based approaches have become popular mostly due to their low data demands and the fact that they bypass parallel corpora. However, joint learning with parallel data (or hybrid approaches which combine mapping-based and joint learning) are very competitive, see, e.g.:
- https://openreview.net/pdf?id=S1l-C0NtwS
- https://arxiv.org/pdf/1906.05407.pdf

===============
Strengths
===============

*The paper is nicely motivated for most parts (although it should stress much more why we require mapping-based approaches and static CLWEs to begin with) and is easy to read.
*The build-up to introducing the actual spectral normalization method makes it easy to trace how the authors thought about the problem, and the insights (or the lack of them) from prior work.
*The method is quite simple to apply and seems widely applicable to any mapping-based CLWE induction method, yield to consistent gains (which do vary across different tasks, but it is always beneficial to apply the normalization method).
*The experiments (when it comes to the focus on static CLWEs) are comprehensive and executed in a sound way.

===============
Weaknesses
===============

*The paper is mostly a continuation of similar work in the same subfield, applying and leveraging insights made by other researchers - e.g., it is heavily based on the work of Dubossarsky et al. (EMNLP-20). There doesn't seem to be enough substance for a full publication.
*Given the results, I am still wondering whether any practitioner will apply this method in practice - also, does the method help with more distant language pairs?
*The paper also relies on some arbitrary choices (the criterion for zeroing-out singular values; the hparam beta) - some baseline methods like the simple iterative-norm method of Zhang et al. are non-parametric, so perhaps they are already in disadvantage in empirical comparisons? Is there any true impact of doing multiple iterations?
*The paper focuses only on static CLWEs, which also limits the impact of this work - since the work is largely empirical, it is unclear who will use the main outcomes of the work.

===============
Quality and clarity
===============

It is very clear what has been done, and the experimental setup along with the description of the core method is very clear as well. My main criticism does not refer to the style of the paper (which is very good), but is more concerned with its novelty, impact, comparisons to (relevant) baselines (see above).

===============
Impact and Significance
===============

As mentioned, the paper covers one niche subfield of learning static CLWEs (with a particular suite of models), and given that it focuses on the paradigm which is now largely surpassed, I am not sure that the paper will have any major impact. It might be cited as another normalization method, but unless a stronger theoretical and empirical motivation is provided, as mentioned before in my review - it is unclear who will use the main outcomes of the work.

**Time Spent Reviewing:**

2.5

---

> ### Author Response · Authors · 2021-08-10
> **cross-validation**
>
> Thank you for the thorough review.  Here are some points not directly address in the global response:
>
> "
> The paper also relies on some arbitrary choices (the criterion for zeroing-out singular values; the hparam beta) - some baseline methods like the simple iterative-norm method of Zhang et al. are non-parametric, so perhaps they are already in disadvantage in empirical comparisons? Is there any true impact of doing multiple iterations?
> "
>
> We used a training set to choose these parameters, and showed they were effective on a test set.  As we carefully followed proper cross-validation methodologies, we do not feel these choices were arbitrary, and the improvements on these choices are born out in the results tables.
> Implicitly, the Zhang et al work uses a very large beta value.

---

### Author Response · Authors · 2021-08-10
**Summary comment about concerns**

We thank the reviewers for their positive and helpful reviews.
We will address all minor comments and incorporate and cite the relevant literature they have mentioned.
Here allow us to discuss the few main concerns about the paper:


Concern 1:  Studying alignment of non-contextual embeddings is waning.

In recent times, contextual embeddings (e.g., BERT) have demonstrated significant improvement in tasks compared to non-contextual ones. However, for tasks that study the typography of languages, static word embeddings are mostly used. These static embeddings also have their place in NLP.
  - Contextual embeddings, even more than non-contextual ones, relies on enormous training data to train the more complex models.  This puts low-resource languages at an even further disadvantage even though these BERT models have shown great yields in the area of cross-lingual transfer from high resource languages to low resource languages.
  - There are numerous applications for embeddings outside of NLP (images, graphs, spatial data, merchants, biology) where there is no concept of contextual embedding. The attempts to generalize this idea have not surpassed the standard ones.  It may be useful to have data trained from separate contexts combined into the same joint embedding in these settings.  We did not discuss these applications as much because of the preponderance of work in this area, including many recent papers focusing on NLP applications.  This allows us to have more direct comparisons and benchmarks.
  - Despite these challenges, our work can potentially be extended to contextual embeddings (https://arxiv.org/abs/2107.09186). This paper was public after we submitted our work.



Concern 1a:  Cross-lingual embeddings do not need to be aligned if they can be jointly trained.

The second aspect of the first concern is the alignment issue, as opposed to the potential issue of studying non-contextual embeddings.
   - The optimization involved in most embeddings is inherently at odds with the multilingual settings.  The model used to train these embedding attempts to predict a missing word from context, but since the trained data is almost entirely composed of mono-lingual elements, the prediction does (and should) bias towards keeping words from the same language similar and other languages apart.   Whereas embedding languages separately and then aligning the languages does not have this same concern.  The joint-embedding results are typically only competitive (in BLI score) with embed then align scores when an alignment step is applied after the joint alignment anyways.
  - Joint training can be far more complicated and expensive than the simple alignment step we study.  Although the empirical evaluation is time/compute-intensive, our alignment step is quite simple and efficient.  On the other hand, these training embeddings can require hundreds of thousands of dollars of computing time, which has well-documented negative consequences for the planet and accessibility of the research outside of compute-rich research labs and companies.



Concern 2:  Some experimental baselines are missing.

We did compare our results extensively to other baselines like (CCA (https://aclanthology.org/E14-1049.pdf), PROC (https://arxiv.org/pdf/1710.04087.pdf), DLV (https://arxiv.org/pdf/1808.09334.pdf), PROC-B (https://arxiv.org/pdf/1902.00508.pdf), RCSLS (https://aclanthology.org/D18-1330.pdf), and VecMap (https://aclanthology.org/P18-1073.pdf)) in table 4 (also see table 15 and 16 in the supplementary Supplementary Material).  The main difference between the papers the reviewers provided as the baseline is that the evaluation metric for the bilingual lexicon induction (BLI) task was Precision at 1 (P@1). In contrast, we used mean average precision (MAP) equivalent to mean reciprocal rank (MRR). P@k treats all models that rank the correct translation below k equally (https://arxiv.org/pdf/1902.00508.pdf) were as MAP is more informative.



Concern 3:  This paper is too "thin"  or it just consolidates research.

We believe the consolidation of research, and identifying several effective approaches as types of normalization, and then combining them into a single improved approach, is an important observation.  While we believe our method is the final and "right" way to accomplish this, our paper also identifies normalization as an orthogonal component to the alignment and measurement methodology, and would allow another researcher to determine if it can be improved without entangling the issue with separate elements.

---

### Decision · Program_Chairs · 2021-09-27

**Decision:**

Reject

**Comment:**

This paper proposes a technique for adjusting the spectral properties of word embeddings that is used to make cross-lingual alignment more straightforward. Experiments show that the proposed method results in better cross-lingual generalisation and bilingual lexicon induction. Overall the method and results are sound, but the normalisation trick is found to be a bit limited as a contribution for this paper, especially given the large number of papers that have appeared related to this issue over the last couple of years (adjusting spectral properties of word embeddings, improving bilingual alignment, etc). Finally, all reviewers remark that this is a well-written, clear paper- in conjunction with a bit more content, it would be a much stronger contribution.